# Copper-Catalyzed Reactions of Aryl Halides with *N*-Nucleophiles and Their Possible Application for Degradation of Halogenated Aromatic Contaminants

Tomáš Weidlich [1],*, Martina Špryncová [2] and Alexander Čegan [2]

[1] Chemical Technology Group, Institute of Environmental and Chemical Engineering, Faculty of Chemical Technology, University of Pardubice, Studentská 573, 532 10 Pardubice, Czech Republic
[2] Biochemistry Group, Department of Biological and Biochemistry Sciences, Faculty of Chemical Technology, University of Pardubice, Studentská 573, 532 10 Pardubice, Czech Republic
* Correspondence: tomas.weidlich@upce.cz; Tel.: +420-46-603-8049

**Abstract:** This review summarizes recent applications of copper or copper-based compounds as a nonprecious metal catalyst in N-nucleophiles-based dehalogenation (DH) reactions of halogenated aromatic compounds (Ar-Xs). Cu-catalyzed DH enables the production of corresponding nonhalogenated aromatic products (Ar-Nu), which are much more biodegradable and can be mineralized during aerobic wastewater treatment or which are principally further applicable. Based on available knowledge, the developed Cu-based DH methods enable the utilization of amines for effective cleavage of aryl-halogen bonds in organic solvents or even in an aqueous solution.

**Keywords:** amine; amination; arylation; halogenated aromatic compounds; aryl halide; dehalogenation; C-N cross-coupling



## 1. Introduction

Ar-Xs are technologically important inert low-cost solvents (chlorobenzene or o-dichlorobenzene) and intermediates for the manufacture of flame-retardant polymers used in electronics and furniture. Furthermore, Ar-Xs are the chemicals necessary for the production of industrially important dyes, pigments, and a broad group of biologically active species such as pesticides and drugs. Ar-Xs are even directly used as biocides (2,4,6-tribromophenol used as a wood preservative or fungicide, Triclosan or Triclocarban used as antibacterial agents, chlorhexidine or chloroxylenol used as disinfectants, etc. [1]). On the other hand, Ar-Xs are common xenobiotics resistant to biodegradation, often exhibit considerable toxicity, and have long been regarded as a significant source of environmental pollution [2]. The presence of Ar-Xs in effluent discharges is of increasing interest due to the ecological effects and possible negative impact on public health [2].

Utilization of Ar-Xs as arylating agents based on Cu-catalyzed substitution of bound halogen (X) applying different nucleophiles serves as the technique for production of a broad scale of useful chemicals such as aromatic amines, ethers, phenols or sulfides, and alkylated or arylated aromatic compounds via Ullmann and Ullmann-like reactions [3,4]:

$$Ar\text{-}X + NuH \rightarrow Ar\text{-}Nu + HX; NuH = \text{nucleophiles}$$

Besides the above-mentioned, the substitution of halogen(s) bound in Ar-Xs serves as an effective treatment method for degradation of low polar and persistent Ar-Xs, converting them to Ar-Nu, non-halogenated and commonly more biodegradable and less toxic products (Scheme 1). In addition, completely dehalogenated compounds are suitable as a high-quality source of energy because they are not precursors for the formation of toxic polychlorinated biphenyls, dibenzo-p-dioxins and respective polyhalogenated dibenzofurans (PCDD/Fs) during incineration (Scheme 1, undesirable reaction pathways) [2]. The

supposed source of waste aryl halides for described dehalogenation typically represents distillation residues from organic fine chemical production sites or halogenated residues produced by electronic waste recycling. Inorganic halides soluble in water are the sole non-toxic by-products of this process.

**Scheme 1.** The proposed addition of N-nucleophiles could significantly minimize the formation of highly toxic halogenated PCDD/Fs-like products during the dehalogenation of Ar-Xs.

Abundant nonprecious transition metals such as copper exhibit interesting catalytic activity in the activation of Ar-Xs for $C_{sp2}$-X cleavage accompanied by substituting the halogen [3–10]. This is permitted due to the easily accessible and reasonable stability of Cu(0), Cu(I), Cu(II) and Cu(III) oxidation states. Mainly Cu(I) salts are used as the sources of active Cu-based catalysts formed in situ during the co-action of auxiliary ligands [3–10].

Recent developments in the area of copper-catalyzed C-O cross-coupling and C-C homo-coupling reactions of Ar-Xs producing biaryls, ethers and phenols were described in the previous review [4]. The Cu-catalyzed conversion of Ar-Xs to biaryls, phenols or aryl ethers is an effective method for $C_{sp}^2$-X bond cleavage; however, it could be an incidental source of undesirable halogenated biphenyls and PCDD/Fs [5], Scheme 1 (undesirable reaction pathways).

The simplest solution for minimizing the risk of potential PCDD/Fs formation during $C_{sp}^2$-X substitutions, caused by the action of O-nucleophiles, is the utilization of another effective nucleophilic agent(s). This paper highlights recent developments for Ar-X amina-

tion based on Cu-catalyzed nucleophilic substitutions published from 2000 to the end of 2021 and abstracted by Web of Science.

This review is focused exclusively on the area of the potential application of metal catalysts based on cheap, low toxic and abundant copper for dehalogenation (DH) via facile aryl-halogen bond scission caused by N-based nucleophilic displacements (Scheme 2). Therefore, this review does not describe tandem reactions comprising C-X cross-couplings with partitions of adjacent groups in subsequent cyclization reactions.

**Scheme 2.** Scope of this work.

## 2. History and Modern Trends in Cu-Catalyzed Ar-Xs Transformations with Amines

Ar-X-based N-arylation of organic amines or amides (Ullmann-Goldberg reactions) is a widely used transformation in chemical synthesis because it allows access to a wide group of dyes, pigments, and biologically active and/or pharmacologically significant compounds [6–9].

In 1906, Irma Goldberg published an arylation of aniline with 2-bromobenzoic acid catalyzed by copper [10], Scheme 3.

**Scheme 3.** Cu-catalyzed debromination of 2-bromobenzoic acid accompanied by arylation of aniline [10].

Such copper-mediated cross-coupling reactions have many industrial applications, including the synthesis of intermediates and fine chemicals for pharmaceutical and polymer chemistry. However, Cu-catalyzed couplings have not been employed to their full potential for a long time.

Until 2000, the main drawbacks of Cu-catalyzed Ullmann-type C-X coupling reactions between Ar-Xs and nucleophiles were the harsh reaction conditions (several hours at temperature as high as 210 °C), the need for stoichiometric use of copper or its salts and the utilization of polar solvents. According to the requirements of organic chemists, the Cu-catalyzed C-N couplings had poor functional group compatibility and poor reaction efficiency. As a result, Pd-based catalysis achieved major development enabling C-N couplings at even ambient temperature using a catalytic quantity of Pd-based catalyst [11–13].

However, taking into account the price of noble palladium and its toxicity, abundant, cheaper and more sustainable C-N coupling catalysts are being sought, such as Ni- or Cu-based catalytic systems [14–22].

Both radical and ionic processes were proposed or even proved for Cu-based C-N coupling catalysis involving the effects of Cu(I), Cu(II) and even Cu(III) oxidation states [3,14,16,22].

The mechanism of Cu-catalyzed amination of Ar-Xs is explained most often by oxidative addition (OA) of Ar-X into the LCu(I)-Nu complex producing LCu(III)XAr intermediate. This LCu(III)XAr decomposes via reductive elimination (RE) producing Ar-Nu and LCu(I)X [22], Scheme 4.

**Scheme 4.** Scheme of radical (SET) and ionic (OA/RE) mechanisms proved for Cu-catalyzed C-N cross-coupling [3,14,16,22].

The third, alternative reaction mechanism is based on the nucleophilic aromatic substitution of halogen in π-complex formed from Cu-ligand and Ar-X [23], Scheme 5.

**Scheme 5.** Possible activation of Ar-X by Cu(I)L discussed by Zhang [23].

Modern trends in organic chemistry, such as the sustainable chemistry principles, motivate more environmentally friendly methodologies based on the application of catalytic amounts of copper catalyst. This requirement was fulfilled mainly by applying auxiliary ligands for control of the coordination environment of the used Cu-species [24–26].

Recently, "ligand free" reaction conditions have been mentioned for the C-N cross-couplings catalyzed by Cu-based compounds [14].

However, with high probability, a used solvent or base may act as a spare ligand in these cases. Guo et al. and Choudary et al. supposed that using a combination of Cu(I) source and $K_3PO_4$ as the base or calcium phosphate as the support, the phosphate group is able to chelate Cu(I) which subsequently assisted the oxidative addition to the Cu center [27,28].

Monnier and Taillefer adverted to lower the reproducibility of such "ligand-free" reaction systems in particular by applying scale-up compared with ligand-based reaction systems [14].

Next to used ligands, the solvents have a significant role to play in the overall sustainability of the chemical processes [29].

Green pathways involving reactions performed in biobased renewable solvents, non-volatile ionic liquids, or polyethylene glycols were utilized to replace the harmful, sometimes even carcinogenic, volatile organic solvents produced from crude oil [29–31].

Dimethyl sulfoxide (DMSO) is a well-known non-toxic polar aprotic solvent produced from a by-product of the papermaking industry, dimethyl sulphide. DMSO is a green replacement for harmful amides such as N-methylpyrrolidone (NMP) or N,N-dimethylformamide (DMF) used as common polar aprotic solvents [32,33].

However, it must be mentioned that the application of DMSO is not compatible with a variety of copper salts, halides or bases at high temperatures which can cause decomposition of DMSO and pose a potential explosion hazard [34].

### 2.1. Cu-Catalyzed Substitution of Halogen in Ar-Xs with NH$_3$

Simple amination of aryl halides using ammonia catalyzed by Cu(I) was extensively studied using polar aprotic (DMF or DMSO) or polar protic (EtOH, ethylene glycol, polyethylene glycol) solvents, different bases (Cs$_2$CO$_3$, K$_2$CO$_3$, K$_3$PO$_4$) and a broad spectrum of ligands (Figure 1) [35–37].

The published Cu-catalyzed methods applied NH$_3$, NH$_4$Cl, NH$_4$OH, acetamidine, amices or amino acids as a source of nucleophiles, in addition to different types of ligands (Figure 1) or even ligandless procedures [38–53], see Scheme 6 and Tables 1 and 2. In most cases, only aryl bromides and iodides are applicable for Cu amination. Aryl chlorides are quite inert toward this type of Cu-catalyzed cross-couplings in most cases [35–44]. The most effective oxamide-based ligands, N-(naphthalen-1-yl)-N-alkyl oxalic acid diamides (MNFMO or NFMO), were found to achieve high turnovers (complete C-N cross-coupling with only 0.1 mol.% Cu$_2$O and ligand) [44]. These ligands achieved 10,000 turnovers in cases of cross-coupling aryl bromides and iodides with ammonia [44].

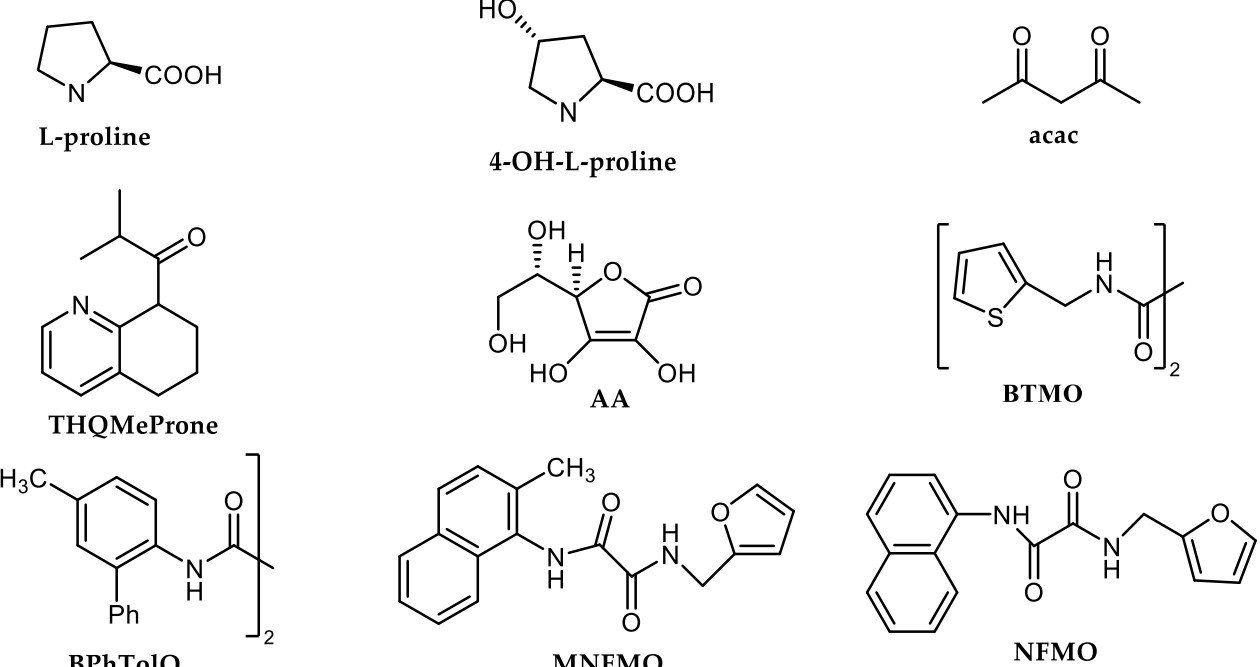

**Figure 1.** Structures of ligands applicable for converting aryl halides to the corresponding anilines [35–37,44–46,50].

**Scheme 6.** Amination of aryl halides using $NH_3$ or its different sources [35–46].

**Table 1.** Cu-catalyzed amination of (hetero)aryl halides using $NH_3$, $NH_4Cl$, acetamidine or valine.

| (Hetero)Aryl Halide (Ar-X) | Added Catalyst | Added Ligand (mol.%) | Base/Solvent | Reaction Conditions | Yield (%) | Ref. |
|---|---|---|---|---|---|---|
| Bromopyridines or iodobenzene | $Cu_2O$ or Cu/CuCl (0.5 wt.%) | no | 8 M $NH_3$ in $HOCH_2CH_2OH$ | 80 °C/16 h in autoclave flushed Ar | 62–99 | [38] |
| Subst. 2-bromopyridines | $Cu_2O$ (2 mol.%) | no | $HOCH_2CH_2OH$ saturated with $NH_3$ | 100 °C/24 h in autoclave | 54–82 | [39] |
| 4-bromo-acetophenone | CuI (equimolar quantity to substrate) or Cu powder (20 mol.%) | no | 27 wt.% $NH_3$ in $H_2O$ + 6 M $NH_3$ in $HOCH_2CH_2OH$ (3:5) | flushed with $N_2$ 85 °C/8–12 h (0.3–1.2 MPa) | 77–86 | [40] |
| Subst. Bromo- and iodobenzenes (chlorobenzenes do not react) | CuI (equimolar quantity to substrate) + Cu powder (20 mol.%) | no | 27 wt.% $NH_3$ in $H_2O$ + 6 M $NH_3$ in $HOCH_2CH_2OH$ (3:5) | flushed with $N_2$ 50–85 °C/8–16 h (ambient pressure) | 37–85 | [40] |
| 3- and 4-subst. Iodobenzenes (2-iodo- with low conversion) | CuI (10 mol.%) | L-proline (20 mol.%) | 1 eq. $NH_4Cl$ + 3 eq. $K_2CO_3$ DMSO + 5 vol.% $H_2O$ | Under Ar 25 °C/12 h | 32–98 | [48] |
| 3- and 4-subst. iodobenzenes (2-iodo- with low conversion) | CuI (20 mol.%) | L-proline (40 mol.%) | 1.5 eq. 28% aq. $NH_4OH$ + 3 eq. $K_2CO_3$ + DMSO | Under Ar 25 °C/24 h | 77–97 | [48] |
| Iodoanilines | CuI (10 mol.%) | L-proline (20 mol.%) | 1.2 eq. of acetamidine.HCl 2–3 eq. $Cs_2CO_3$ In DMF | Under $N_2$ 110–120 °C/10 h | 64–92 | [49] |
| Subst. bromo- and iodobenzenes (Ar-Cls do not react) | CuI (20 mol.%) | no | 1.2 eq. of valine 1.5 eq. $Cs_2CO_3$ in DMSO | (a) 90 °C/24 h under Ar (b) 90 °C/24 h under $O_2$ | 28–90 | [41] |

Simple "ligand-free" protocols for converting different aryl bromides or aryl iodides to the corresponding anilines were published in a paper by Guo et al. and utilized powdered copper or CuI, mixed and heated with ammoniacal aqueous ethylene glycol [40]. As proved, ethylene glycol functions as both solvent and ligand for the in situ formation of active Cu-catalyst [40]. However, this amination was not observed with aryl chlorides (Table 1).

Using the appropriate oxalic acid diamides such as BPhTolO (Figure 1) in 5 mol.% loading together with 5 mol.% CuI, even the amination of non-activated aryl chlorides takes place at higher temperatures in DMSO (Table 2) [45,46].

The mechanism of these highly active oxalic acid diamides was studied by Morarji and Gurjar [3]. Based on DFT calculations and UV-VIS/cyclic voltammetry measurements, the mechanism of Ar-Cls amination catalyzed by Cu-oxalamides is not based on the most

often mentioned oxidative addition of Ar-X into the LCu(I)-Nu with subsequent reductive elimination producing Ar-Nu and LCu(I)X.

**Table 2.** Cu-catalyzed amination of (hetero)aryl halides using $NH_3$ or $NH_4OH$.

| (Hetero)Aryl Halide (Ar-X) | Added Catalyst | Added Ligand (mol.%) | Base/Solvent | Reaction Conditions | Yield (%) | Ref. |
|---|---|---|---|---|---|---|
| Subst. bromobenzenes | CuI (20 mol.%) | 4-OH-L-proline (40 mol.%) | aq. $NH_4OH$ + DMSO (1:2) | 50 °C/24 h under $N_2$ | 55–92 | [50] |
| Subst. bromo- and iodobenzenes | Cu(acac)$_2$ (10 mol.%) | Acac (40 mol.%) | 2 eq. $Cs_2CO_3$ aq.$NH_4OH$/DMF (0.6:4) | 70–90 °C/24 h under $N_2$ | 23–99 (23% ortho-subst.) | [51] |
| Subst. Ph-Br (chlorobenzenes do not react) | CuBr (10 mol.%) | THQMeProne (20 mol.%) | 2.5 eq. $K_3PO_4$ 5 eq. aq.$NH_4OH$ in DMSO | 110 °C/24 h under Ar | 52–95 (52% ortho) | [52] |
| Subst. iodobenzenes | CuBr (5 mol.%) | THQMeProne (10 mol.%) | 2.5 eq. $K_3PO_4$ 5 eq. aq.$NH_4OH$ in DMSO | 25 °C/24 h under Ar | 27–95 (27% ortho) | [52] |
| Brominated N-heterocycles | Cu(acac)$_2$ (5 mol.%) | no | 1 eq. $K_3PO_4$ 20 eq. $NH_3$/DMF | 90 °C/24 h under $N_2$ | 48–88 | [42] |
| Subst. bromo- and iodobenzenes, $NO_2$-chloro-benzenes | CuI (1 mol.%) | AA (1 mol.%) | $NH_3$ (l) | 100 °C/18 h in autoclave | 63–99 | [53] |
| Bromobenzenes | CuI (10 mol.%) | DMEDA (15 mol.%) | 27 wt.% $NH_3$ in $H_2O$/DMSO (3:1) | 130 °C/6–18 h in autoclave flushed Ar | 84–96 | [43] |
| Bromobenzenes | CuI (10 mol.%) | no | 27 wt.% $NH_3$ in $H_2O$/PEG300 (3:1) | 130 °C/12–24 h in autoclave flushed Ar | 85–99 | [43] |
| Subst. bromo- and iodobenzenes | Cu$_2$O (0.1 mol.%) | Ar-Br: MNFMO (0.1 mol.%) Ar-I: NFMO (0.1 mol.%) | 1.3 eq. KOH + 27 wt.% $NH_3$ in $H_2O$ + EtOH (2:1) | Ar-Br: 80 °C/24 h Ar-I: 60 °C/24 h Under Ar | 64–98 | [44] |
| Subst. Chlorobenzenes and chlorinated heterocycles | CuI (5 mol.%) | BPhTolO (5 mol.%) | 1.1 eq. $K_3PO_4$ 2 eq. aq.$NH_4OH$ in DMS | 110–120 °C 24 h Under Ar | 60–95 | [45] |

According to their findings, based on in situ FTIR and [1]H NMR measurements Cu(I) coordinates through both carbonyls from oxalamide, and the corresponding copper complex LCu(I)Nu is the most favorable intermediate of this C-N coupling proceeding via outer-sphere single electron transfer (SET) pathway (Schemes 7 and 8).

**Scheme 7.** Proposed single electron transfer (SET) mechanism for arylation catalyzed Cu(I)/oxamides [3].

**Scheme 8.** Proposed catalytically active Cu(I) species taking part in the amination of Ar-Cls using oxalic acid diamide BTMPO [3].

Similarly, using primary amides of general formula $R^1CONH_2$ and $Cu_2O$/BTMO, effective arylation with aryl chlorides is available [47], Scheme 9.

**Scheme 9.** Arylation of $NH_3$ and amides using aryl chlorides catalyzed by Cu(I)/oxalamide complexes [45,47].

Instead of a harmful polar aprotic solvent such as DMF or toxic polar protic ethylene glycol, more sustainable biobased DMSO or low-toxicity alcohols such as ethanol or polyethylene glycols (PEGs) were used in many cases together with cheap bases such as $K_3PO_4$, KOH or $K_2CO_3$ in published methods (Tables 1 and 2). The most expensive ingredient, auxiliary ligand, seems to still be an essential component for efficient conversion of tested aryl halides including aryl chlorides to the corresponding anilines in most cases. The possible recycling of used ligands, especially those based on oxalic acid diamides, is necessary for the potentially broad application of this type of dehalogenation method.

### 2.2. Cu-Catalyzed Substitution of Halogen in Ar-Xs with Primary and Secondary Amines

Instead of ammonia and its mentioned substitutes, a broad range of other nitrogen nucleophiles including aliphatic and aromatic amines, N-heterocycles, or amino acids can be used for the amination of Ar-Xs (Scheme 10). Interestingly, it was observed that amides

and electron-rich azoles (pyrrole, imidazole or pyrazole) are more reactive than other amine substrates [54,55]. The observed increased reactivity of azoles and amides may be due to the faster reaction rates of Cu-azolate or Cu-amidate complexes in oxidative additions of Ar-Xs compared to other Cu-amino complexes, or due to the higher acidity of azoles and amides compared with other aliphatic or aromatic amines. On the other hand, more acidic polyazoles such as tetrazole exhibit low reactivity toward Cu-catalyzed C-N cross-coupling, likely due to their distinctive acidity and low N-nucleophilicity [54].

**Scheme 10.** Arylation of different primary or secondary amines [6–10,14–28,46,51,54–71].

Buchwald's research group developed polyamine-based ligands such as 1,2-diamines (N,N′-dimethylethane-1,2-diamine (DMEDA) or trans-N,N′-dimethylcyclohexane-1,2-diamine, DMCHDA) and (substituted) phenanthroline [56–59], Figure 2, Scheme 10.

**Figure 2.** N,N-bidentate nucleophiles effective for Cu-catalyzed C-N cross-coupling [22,56–59].

DMEDA or DMCHDA were utilized together with $CuI/K_3PO4$ for effective arylation of indoles using Ar-I and Ar-Br in boiling toluene [57].

In particular, DMCHDA was recognized as a very powerful ligand for arylation of azoles and diazoles in boiling toluene using $CuI/K_3PO_4$ as the commonly available reagents and applying Ar-I or even Ar-Br [56,57]. Aryl bromides Ar-Br were used for arylation of indazoles. However, the regioselectivity of N-1 arylation was significantly lower due to the slow oxidative addition of Ar-Br to the Cu-indazole complex, which is rearranged from the initially formed N-1 to the N-2 regioisomer [56]. Authors applied halide exchange protocols for converting Ar-Br to Ar-I [60–62] by using the action of NaI/CuI/diamine in boiling toluene for straightforward arylation of indazoles.

4,7-Dimethoxy-1,10-phenanthroline (DiMeOphen, Figure 2) was developed as a superior ligand for arylation of substituted imidazoles and benzimidazoles under relatively mild conditions [58,59].

Several in situ-produced enamine, oxime and hydrazone-based Cu(I) complexes (Figure 3) were described as highly effective for reactions between Ar-Xs and different electron-reach azoles even in boiling acetonitrile in co-action of $Cs_2CO_3$ at approximately 82 °C [63–65]. Additionally, salicylaldehyde-based oxime (SAO) or hydrazide (SAH) are simply available and cheap ligands that produce air-stable Cu(I)complexes and catalyze smoothly-described reactions of Ar-I and Ar-Br substituted with both electron-donating and electron-withdrawing functional groups.

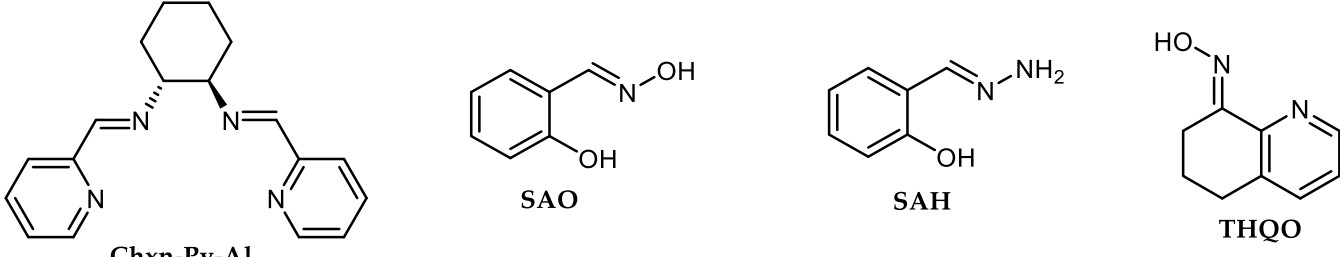

**Figure 3.** Imine, oxime or hydrazone-based ligands applicable for the production of air-stable Cu-catalysts [14,35,46,63–65].

Some amino acids such as proline or N,N-dimethylglycine were proved as effective bidentate auxiliary ligands for Cu-catalyzed C-N cross-couplings [66,67]. Besides the above-mentioned, amino acids possessing the primary amino group are simply arylated using both Ar-I and Ar-Br [65,66,68].

Generally, alkyl amines are more reactive in Cu-catalyzed N-arylation than anilines due to the stronger coordinating ability of the alkyl amine nitrogen compared to that of the aniline [23,54,69]. Using L-proline or N,N-dimethylglycine as the ligands, arylation of alkyl amines occurred at significantly lower temperatures compared with arylation of anilines [23].

Even at room temperature, the Cu-based C-N couplings were observed. Shafir and Buchwald described smooth C-N coupling of Ar-I and primary along with several secondary amines catalyzed with Cu(I) complexes produced in situ from CuI CuI (5%) and oxime-based ligand THQO, or 1,3-diketone-derived ligands CHXMK and CHXPrK (Figure 4). The mentioned room-temperature C-N coupling was performed in DMF using $Cs_2CO_3$ base within 2–4 h time period [69]. However, using aryl bromides, above mentioned arylation proceeds at 90 °C. Ar-Cls are inert toward tested amination. Two 1,3-dicarbonyl compounds-based ligands such as EtOCHXCARB or DPyPDON (Figure 4) were recognized as effective for Cu-catalyzed cross-coupling of different aromatic N-heterocycles and cyclic amide (pyrrolidone) with aryl iodides at mild reaction conditions [70,71]. Xi et al. prepared a catalytically active Cu-based complex via in situ reaction of 1,3-diketone DPyPDON and CuI for arylation of imidazoles by aryl bromides [70].

**CHXMK: R = CH₃**
**CHXPrK: R = CH(CH₃)₂**

**EtOCHXCARB**

**DPyPDON**

**Figure 4.** Structures of highly active 1,3-diketone-based ligands [69–71].

Cross-coupling of bulky aliphatic amine-based cross-partners was observed by Cook's group using pyrrole-ol ligand stabilized by Hantzsch ester towards rapid degradation [72], Scheme 11.

Usually, the inertness of aryl chlorides towards Cu-catalyzed N-arylations is the main limitation in the application of cheap Ar-Cls. The important exception to the inertness of Ar-Cls is a group of Ar-Cls activated by substitution with electron-withdrawing group(s) (Ewg). Mentioned chloroaromatics substituted with Ewg, especially chloronitrobenzenes, are activated for arylation of nucleophiles via $S_NAr2$ reaction mechanism and react with amines even under catalyst-free conditions, Scheme 12 [73].

**Scheme 11.** Arylation of sterically hindered amines using even ortho-substituted Ar-I catalyzed by Cu/pyrolle-ol [72].

**Scheme 12.** Arylation of nucleophile via $S_N$Ar2 mechanism (the non-aromatic intermediate is stabilized by Ewg [73].

In addition, 2-Chlorobenzoic is another well-known and often applied Ar-Cl-based arylating agent containing a carboxyl group in the role of Ewg. However, 2-chlorobenzoic acid requires the application of a Cu-based catalyst for effective C-N cross-coupling [74–77].

Attempts were made to utilize effective bidentate ligands for arylation using non-activated aryl chlorides (chlorobenzene, chlorotoluene, etc.) under vigorous conditions in high-boiling polar aprotic solvents or in excess Ar-Cl used as both reagent and solvent.

Piperidine-2-carboxylic acid L-PIPA (Figure 5) was detected as a low-cost N,O-bidentate ligand applicable for amination, respective amidation of aryl chlorides in co-action of CuI in hot $K_2CO_3$/DMF mixture, although resulting in a low yield of corresponding anilides or anilines [78] (Scheme 13). Aniline and nitroaniline regioisomers were tested for amination of chlorobenzene. Surprisingly, diphenyl amine was obtained in the lowest yield (15%), 2-nitroaniline as the most reactive amine produces 31% of 2-nitrodiphenyl amine. Arylation of indole was tested with a comparable yield of 36%. Using acetamide, acetophenone was isolated in 24% yield. AO/RE mechanism was proposed for this reaction.

N,N′-Dimethylcyclohexane-1,2-diamine DMCHDA (Figure 2) was proved to be an effective ligand for amidation of aryl chlorides used in excess as both arylation agents and solvent with the addition of CuI and $K_3PO_4$ at 130°C after tens of hours [79,80], Scheme 13.

**L-PIPA**          **Oxine-N-oxide**          **MEAPYO**

**Figure 5.** Structures of ligands capable together with Cu(I) to catalyze amination of Ar-Cls [78,81,82].

Furthermore, 8-Hydroxyquinoline-N-oxide (Oxine-N-oxide, Figure 5) was described as a useful O,O-bidentate ligand for effective Cu-catalyzed amination of aryl chlorides using different primary and secondary aliphatic amines and five-membered N-heterocycles including pyrrole, imidazole, pyrazole, indole, 1,2,4-triazole [81], Scheme 13, Figure 5. The AO/RE mechanism depicted in Scheme 4 was suggested for this reaction.



**Scheme 13.** Published amination or amidation of Ar-Cls using different bidentate ligands and CuI [78–82].

Recently, a new complex produced in situ from 2-mesitylamino pyridine-1-oxide (MEAPYO, Figure 5) and CuI was discovered as an available catalyst for efficient amination of aliphatic primary and secondary amines [82]. Liu et al. described CuI/MEAPYO as effective even for C-N coupling of nonactivated aryl chlorides [82]. Using primary amines or less sterically hindered secondary amines, different aryl chlorides produce corresponding arylated amines using 5–10 mol.% CuI and 5–10 mol.% of MEAPYO under heating at 130 °C in DMF using $Cs_2CO_3$ as the base under argon [82]. However, when using more bulky secondary amines (such as N-benzyl-N-methylamine or N-methylpiperazine), incomplete conversion (and lower yield 53–59%) is observed [82], Scheme 13.

In addition to the above-mentioned, Ma´s research group searched for active ligands based on oxalamide derivatives for N-arylation, based on aryl chlorides application [8,47,83,84]). De et al. assumed that furane and thiofene ring moiety bound in oxalamide skeleton (BFMO and BTMPO ligands, Figure 6) are very effective ligands applicable for C-N couplings between aryl chlorides and different N-nucleophiles such as amides and secondary amines, including heterocyclic ones [47].

Primary and secondary amines react smoothly with aryl bromides and iodides using even highly catalytic (0.1 mol.% $Cu_2O$ and ligand, over 10,000 turnovers) conditions when applying a 1-naphtylamine-based oxalic diamide such as MNBO (N-(2-methylnaphtalen-1-yl)-N′-benzyl oxalamide (Figure 6) in boiling KOH/EtOH mixture under inert after 12 h of action [44].

The weakly-activated polyaromatic chlorides such as 1-chloroanthraquinone react smoothly with aromatic amines so long as a stochiometric quantity of powdered copper is added. This reaction is broadly used in anthraquinone-based dye and pigment production. Zhang et al. improved this methodology using only a catalytic amount of CuI (10 mol.%)

in hot N,N-dimethylformamide using cheap $K_2CO_3$ as the base without the necessity of adding another ligand [85], Scheme 14.

BFMO                    MNBO                    BTMPO

**Figure 6.** Structures of another highly effective oxalic acid diamide- and benzoin oxime-based ligands [44,47,83,84].

**Scheme 14.** Synthesis of anthraquinonoid dyes intermediates via amination of 1-chloroanthraquinone [85].

*2.3. Cu-Catalyzed Functionalization of Ar-Xs in Green Solvents*

The replacement of organic solvents produced from fossil fuel sources with low toxic bio-based (green or sustainable) solvents has become a prime concern due to environmental reasons.

Zhang et al. studied the role of amino acids in promoting CuI-based formation of tertiary amines from Ar-Br or Ar-I and secondary amines, Scheme 15 [23]. The reported synthetic protocol was based on the application of green and bio-based solvent dimethyl sulfoxide (DMSO) applied as an excellent solvent for both inorganic and organic compounds taking part in described aminations. Used amino acids are cheap and simply recycled by washing the evaporated reaction mixture with water, according to the authors [23].

**Scheme 15.** Arylation of amines using amino acid/CuI catalysts in dimethyl sulfoxide [23].

Yuan et al. discovered α-benzoin oxime (BO) as a useful ligand for arylation of a wide of nucleophiles (e.g., azoles, piperidine, pyrrolidine and amino acids) using (hetero)aryl halides in moderate to excellent yields [86], Schemes 16 and 17. The protocol based on the

application of Cu(OAc)$_2$/K$_3$PO$_4$/BO in DMSO allows rapid access to the most common scaffolds found in FDA-approved pharmaceuticals.

**Scheme 16.** Amination of substituted 2-chloropyridines or pyrimidines promoted by Cu(OAc)$_2$/α-benzoin oxime [86].

**Scheme 17.** Amination of substituted bromobenzenes promoted by Cu(OAc)$_2$/α-benzoin oxime [49].

In order to develop a sustainable approach for the construction of C-N cross-coupling, Yadav et al. devised a simple copper-mediated arylation of indoles with aryl halides in glycerol solvent, Scheme 18 [87].

**Scheme 18.** N-arylation of indoles or other five-membered N-heterocycles using CuI/glycerol as recyclable catalyst [87].

The employment of glycerol solvent allowed the simple extraction of products into the diethyl ether and the subsequent efficient recycling of the undissolved glycerol/catalyst layer, after adding fresh DMSO for up to four runs without any loss in the catalytic activity.

Khatri and co-workers used glycerol as a green recyclable solvent to perform a Cu(acac)$_2$ mediated C-N cross-coupling reaction of Ar-I or Ar-Br with various amines, Scheme 19 [88]. The application of KOH and Cu(acac)$_2$ as the cheap, soluble reactants, and the subsequent product isolation by ether extraction in addition to the catalyst recovery becomes possible upon the use of glycerol as the polar protic solvent [88].

**Scheme 19.** Amination of Ar-I or Ar-Br with primary amines or secondary cyclic amines using recyclable Cu(acac)$_2$/glycerol [88].

Bollenbach et al. discovered facile arylation of the primary aliphatic or aromatic amines using Ar-Br affected by Cu(II)salt/glucose/1,3-diketone at 50 °C in nonionic surfactant/water mixture with tBuONa as the base (Scheme 20) [89]. Glucose acts as a reductant of Cu(II) ion, forming catalytically active complex LCu(I)X with 1,3-diketone (dipivaloylmethane) used as the ligand (L). The authors propose the common OA/RE mechanism for this green arylation. The addition of 2 wt.% of nonionic surfactant overcomes the problem with low solubility of Ar-Br in an aqueous solution. The authors did not mention the recyclability of the used catalyst or the nonionic surfactant.

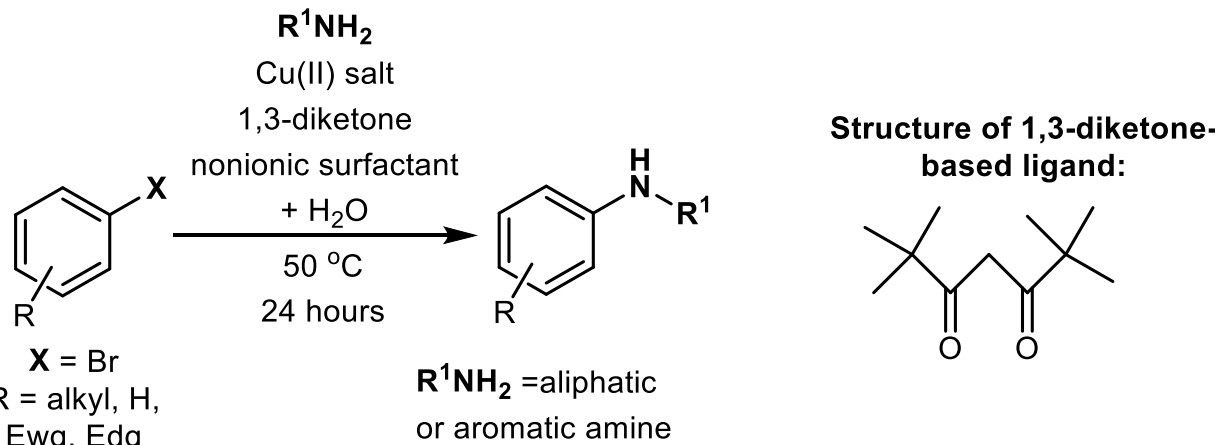

**Scheme 20.** Amination of Ar-Br in aqueous solution using in situ produced LCu(I) catalyst [89].

Arylation of both the aliphatic primary and secondary amines using Ar-I and CuI/oxime THQO (Figure 3) ligand in aqueous KOH solution at 25–65 °C was described by Wang et al., Scheme 21 [80].

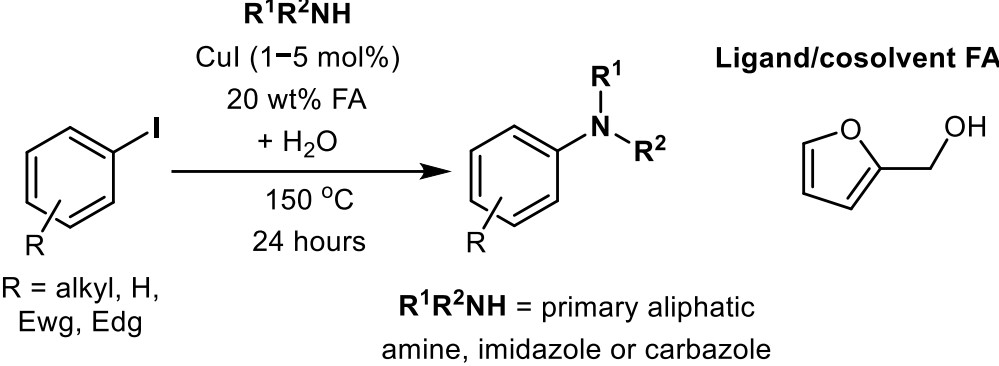

**Scheme 21.** Amination of Ar-I in aqueous KOH solution using CuI/THQO at room or slightly elevated temperature [90].

Ferlin et al. overcame the negligible solubility of Ar-Xs in neat water by using an azeotropic mixture of biomass-derived furfuryl alcohol (FA) and water for effective coupling of Ar-Is with heteroaromatic or aliphatic amines in the presence of CuI/K$_3$PO$_4$ at 150 °C under "ligand-free" conditions, Scheme 22 [91]. FA appears to work as both the solvent and ligand in this case. The authors documented simple removal and recyclability of used FA/water by azeotropic distillation under nitrogen and calculated E-factor 0.97 for this synthetic protocol (E-factor = (kg of waste/kg of product) [92]).

**Scheme 22.** Application of azeotropic mixture furfuryl alcohol/water for arylation of primary aliphatic amines or azoles [91].

*2.4. Microwave Assisted C-N Cross-Couplings*

Generally, the Cu-catalyzed DH reactions were restricted by the harsh reaction conditions and often required high temperatures (100–180 °C) for an extended reaction time. As a result, enormous efforts have been paid to achieve more sustainable reaction conditions by applying alternative energy supplies, such as microwave irradiation. The microwave-based heating dramatically reduces the reaction time required and therefore results in an increase in the DH efficiency [93].

Dihydrazones produced in situ from oxalyldihydrazide and cyclohexanone, acetone, butanone and especially hexane-2,5-dione were published as promising ligands for C-N coupling reactions using CuO in aqueous KOH solution containing phase-transfer catalyst Bu$_4$NBr under conventional or microwave heating [94] (Scheme 23).

This new three-component catalytic system comprising CuO/oxalyldihydrazide/hexane-2,5-dione is suitable for the amine arylation using Ar-I or Ar-Br in an aqueous medium, Scheme 23 [94]. The reaction worked well under microwave irradiation as well as under conventional heating. Several primary and secondary amines, including N-containing heterocycles, were effectively arylated (Scheme 23) [94].

NHR$^1$R$^2$
5 mol% CuO/KOH
50 mol% oxalyldihydrazide
2,5-hexanedione
Bu$_4$NBr/H$_2$O
MW/5 min.
(25−140 °C)

X = I, Br
R = alkyl, aryl,
Ewg, Edg

**Scheme 23.** Microwave-assisted C-N cross-coupling in aqueous KOH [94].

Another ligand-free microwave-assisted N-amidation of indoles and benzimidazoles using Ar-I and a catalytic amount of Cu$_2$O in polyethylene glycol PEG 3400 promoted by Cu$_2$O/Cs$_2$CO$_3$ was published by Colacino et al. [95], Scheme 24. The reaction products are simply isolated without the need for column chromatography by diluting with diethyl ether, aiding in the recovery of insoluble catalyst and the evaporation of ethereal extract. This ligand-free C-N cross-coupling is catalyzed by Cu nanoparticles in situ prepared by microwave heating the Cu$_2$O/PEG 3400/Cs$_2$CO$_3$ mixture. In this case, contrary to the above-mentioned C-N cross-couplings, the authors do not use inertization of the reaction mixture, neither by nitrogen nor by argon.

10 mol% Cu$_2$O
2 eq. Cs$_2$CO$_3$
PEG 3400
microwaves
150 °C/1 h

R = H,
Edg or Ewg
Y = CH or N

**Scheme 24.** Arylation of indole or benzimidazole in polyethylene glycol using microwave heating [95].

Using brominated anthraquinone derivatives such as bromamine acid, the amination produces anthraquinone dye intermediates in an aqueous or alcoholic solution after the addition of Cu powder or Cu(OAc)$_2$ and inorganic bases such as KOAc or alkaline salts of phosphoric acid, especially under microwave heating [96–98], Scheme 25.

Both ionic liquids and deep eutectic solvents were reported as sustainable nonvolatile green solvents applicable for C-N cross-couplings [92,93].

Wu et al. have replaced the commonly used toxic solvents such as DMF and developed a practical method for the CuI mediated arylation of aromatic amines with Ar-Br or Ar-I, involving K$_2$CO$_3$ or t-BuOK as the base in a biodegradable low melting mixture choline chloride/glycerol commonly called deep eutectic solvent (DES), Schemes 26 and 27 [99]. The authors proposed a combined action of both components (choline chloride and glycerol) as ligands in this described "ligand-free" method, comprising an oxidative addition/reductive elimination mechanism. Interestingly, this reaction was performed without inertization in the presence of air at 60–100 °C. In addition, the separation of the product from the reaction mixture is based on a simple extraction using sustainable ether (cyclopentyl methyl ether) and successful repeated recycling of both catalyst and DES has been proved.

**Scheme 25.** Amination of bromoanthraquinone derivatives [96–98].

**Scheme 26.** Possible catalytic pathway of DES promoted the arylation of secondary amines via OA/RE mechanism [99].

**Scheme 27.** Ionic liquid or DES mediated arylation of primary or secondary amines [99,100].

Some task-specific ionic liquids possess good thermal stability, sufficient solubilization ability for both organic and inorganic compounds, and the ability to stabilize $Cu_2O$ nanoparticles. Tetrabutylphosphonium acetate (n-$Bu_4POAc$) was proved as a suitable solvent for simple preparation of nanoscale $Cu_2O$ from $CuCO_3$ and for subsequent arylation of primary and secondary amines using Ar-I without the use of any base under ligand-free conditions on air using nanoscale $Cu_2O$ [100], Scheme 27. The products were separated by simple extraction using alkane. On the other hand, the authors did not choose recyclability of the used ionic liquid/nano $Cu_2O$ mixture.

Lv et al. described IL-proline-butyl-3-methylimidazolium tetrafluoroborate (BMIMBF4) as the recyclable solvent and CuI/L-proline complex as the catalyst for amination of brominated heterocycles and several bromoaromates [101]. Most of the cross-coupling reactions mentioned in Sections 2.2–2.4 require the exclusion of air in most cases. This is not a drawback for the scale-up of Cu-catalyzed dehalogenations, taking into consideration the fact that most of the solvent-based processes in organic technology underwent using inertization of the reaction mixtures for fire hazard minimization.

### 2.5. DH Catalyzed by Reusable Heterogeneous Copper Catalysts

As Cu-based arylation reactions need a high quantity of copper catalyst, its recyclability and reusability are desirable according to the sustainable chemistry principles (minimization of waste production and energy consumption) [102].

The recyclability is achieved mostly using heterogeneous catalysts (nanoparticles (see Scheme 27 [100]), immobilization of catalysts on organic polymeric or porous inorganic supports [103]).

Formation of Cu-complex by reacting CuI with amino acid anions bound in heterogeneous, with imidazolium cations, modified polystyrene is one of the practically proved strategies for enabling the repeated reusability of catalytic systems for the N-arylation of heteroaromatics, Figure 7, Scheme 28 [104].

Chen et al. demonstrated that the recyclability of the supported ionic liquid catalyst up to nine times makes the process green and efficient [104].

Rosa canina fruit extract (usually commercially used as a cosmetic ingredient with CAS No. 84696-47-9, containing namely L-ascorbic acid and polyphenols) was used as a stabilizing and reducing agent for CuO nanoparticles for arylation of N-heterocycles in hot DMF under aerobic conditions [105], Scheme 29. The authors documented that the used nano CuO is recyclable at least six times without loss of activity.

**Figure 7.** Proposed structure of PS-supported MIM.L-proline-based-Cu-catalyst [104].

R = NO₂ or CN and X=Br or Cl
R = Edg or other Ewg; X = Br

**Scheme 28.** Arylation of imidazole catalyzed by CuI/PS-MIM.proline [104].

R = H,CH₃; X=Br or Cl

R¹R²NH = aniline, benzylamine,

indole, imidazole

**Scheme 29.** Arylation of primary and secondary amines using both Ar-Br and Ar-Cl catalyzed by CuO nanoparticles [105].

Another air-stable reusable catalyst based on commercially available CuO nanoparticles (particle size 33 nm and surface area 29 $m^2$/g, Sigma-Aldrich Suppl.) described by Rout et al. catalyzes iodobenzene-based arylation of anilines, benzylamine and other primary or secondary amines in KOH/DMSO (Scheme 30) [106]. The arylation proceeds even using bromo- or chlorobenzene and aniline-yielding diphenylamine in 60% or 80%, respectively.

Nanocrystalline CuO from another supplier, NanoScaleMaterials Inc, with a surface area of 136 $m^2$/g and crystallite size of 7–9 nm, catalyzes arylation of N-heterocycles with activated aryl chlorides and aryl fluorides. Typically, chloro- or fluoro-nitrobenzenes were applied for arylation of imidazole using $K_2CO_3$ as the base and mentioned nanocrystalline CuO in hot DMF (Scheme 31) [107]. The used catalyst was separated by centrifugation and recycled five times without significant loss of activity. However, chlorobenzene is quite inert toward used reaction conditions.

**Scheme 30.** Arylation of amines in DMSO using CuO nanoparticles [106].

**Scheme 31.** Nano-CuO catalyzed arylation of imidazole using activated aryl chlorides or fluorides [107].

Magnetically simply separable silica supported $Cu/Fe_3O_4$ heterogeneous catalyst was prepared by Nasir Baig and Varma [108]. Using this recyclable catalytical system, aryl iodides and activated aryl bromides work as an arylating agent for primary and secondary amines using microwave heating in an aqueous $K_2CO_3$ solution. Chlorobenzene does not react with pyrrolidine under the described reaction conditions [108].

Kore and Pazdera described the preparation of the new stable Cu(I)-based cross-coupling catalyst by ion exchange using polyacrylate katex resin [109]. This polymer supported Cu(I) catalyst enables C-N cross-coupling between 4-chloropyridinium chloride and different amines using $K_2CO_3$ in boiling isopropyl alcohol on air (Scheme 32).

**Scheme 32.** Amination of 4-chloropyridine using Cu(I)-polyacrylate heterogeneous catalyst [109].

Reddy et al. supported in situ reduced copper on cellulose and tested the activity of this heterogeneous catalyst for arylation of N-heterocycles using aryl halides, including aryl chlorides mixture $K_2CO_3/DMSO$ at 130 °C. The catalyst is simply recyclable by filtration several times (Scheme 33) [110].

**Scheme 33.** Arylation of imidazole catalyzed by Cu(0) supported on celulose [110].

CuCl/Fe$_3$O$_4$/polyvinylalcohol-based simply recyclable magnetic nanocatalyst can catalyze C-N cross-coupling even between chlorobenzene and nitrogen heterocycles in Et$_3$N/DMF mixture at 100 °C [111].

1,2-Substituted 1,2-dihydroquinoxaline ligand bound in cross-linked polystyrene (Figure 8) was verified as a simply recyclable source of active C-N cross-coupling catalyst for arylation of aromatic amines with iodo-, bromo- and even chlorobenzene in DMSO, Figure 8 and Scheme 34 [112].

**Figure 8.** Structure of Cu(0)-PS-1,2-dihydroquinoxaline (PS-DHQ-Cu) catalyst [112].

**Scheme 34.** Arylation of anilines or indole by Ar-X [112].

Hydrothermally prepared nano-CuI was proved as an effective recyclable arylation catalyst for C-N cross-coupling between aryl chlorides and primary or secondary amines, including 5-membered N-heterocycles using K$_2$CO$_3$ as the base in hot DMF on air [113], Scheme 35. The used catalyst was repeatedly separated by centrifugation and reused without remarkable loss of activity.

**Scheme 35.** Arylation of primary or secondary amines with aryl chlorides catalyzed by recyclable nano-CuI [113].

Cu-pyridine complex bound on a mesoporous silica SBA-15 surface through melamine connection was proved as an effective and simply recyclable catalyst for arylation of different primary and secondary amines using chlorobenzene in DMF/Et$_3$N on air at 60 °C (Scheme 36 and Figure 9) [114,115].

**Scheme 36.** Facile arylation of anilines, benzylamine and several N-heterocycles catalyzed by CuI/py on SBA-15 supported catalyst [114].

**Figure 9.** CuI/py catalyst immobilized on mesoporous SBA-15 surface (CuI/py on SBA-15) [114].

A comprehensive review dealing with silica supported recyclable Cu-based catalysts is provided by Veerakumar et al. [116].

Nitrogen-rich copolymeric microsheets with molar ratio C/N = 1/2 were prepared through nucleophilic substitution of cyanuric chloride with melamine in pyridine/DMF for supporting and stabilizing $Cu^0$ nanoparticles. These were prepared by impregnation of microsheets with copper acetate and subsequent reduction by hydrazine (Scheme 37). Prepared monodispersed $Cu^0$ nanoparticles were discovered as a superior catalyst for C-N cross-coupling, even aryl chlorides with amines [117].

**Scheme 37.** Preparation of highly active nanoCu$^0$ stabilized in copolymeric carbon nitride CN$_2$ microsheets [117].

Similarly prepared polymeric carbon nitride-supported $Cu^0$ served as a worse C-N cross-coupling catalyst and yielded only 30% arylation using chlorobenzene [118].

Summarizing the above-mentioned, it could be said that the most effective recyclable heterogeneous Cu-based catalysts are applicable even in air, using Ar-Cl as an arylating agent. This opens up possibilities for the use of C-N cross-couplings for the dehalogenation of recalcitrant Ar-X-based waste.

## 3. Conclusions

Utilization of aryl halides for arylation of amines using catalysts based on copper as a cheap and biogenic element potentially enables simple and safe destruction (dehalogenation) of waste non-biodegradable aryl halides to the corresponding aryl amines. Produced aromatic amines could be suitable for subsequent utilization as synthetic intermediates or for energy utilization as the refuse-derived fuel (RDF) [119]. The main advantage of the possible utilization of mentioned C-N cross-coupling reactions compared to C-C or C-O couplings is the minimization of the risk of the potential undesirable formation of highly toxic and thermodynamically stable polyhalogenated biphenyls, dibenzo-p-dioxins, or dibenzofurans as the by-products of arylation reactions [120].

For this purpose, a broad range of bio-based amines such as ammonia from anaerobic digestion of waste biomass or amino acid mixtures (alanine, cysteine, glycine, proline, valine [41], etc.) produced by waste protein hydrolysis are available as a potential source of N- or S-nucleophiles and/or auxiliary ligand(s) [121–123].

Considering that both S- or C-acid-based nucleophiles could be part of waste containing Ar-Xs or should be used as ligands for DH (1,3-diketones); however, their participation in arylation reactions is possible. On the other hand, almost entirely aryl iodides or aryl bromides are necessary for the arylation of sulphides or C-acids [46,124–127].

However, the possibilities of C-N -based multiple cross-couplings of polychlorinated benzenes were never studied in detail according to our best knowledge. Merely differences in cross-couplings of aromates substituted with different halides (typically bromo-iodobenzenes, bromo-chlorobenzenes and halogeno-fluorobenzenes) were mentioned in published articles [59,60,70,128–135].

The broader application of cross-coupling reactions for effective dehalogenation of waste aryl halides should be joined with possible efficient recycling of used catalysts. Cu-based nanoparticles were recognized as reactive enough even for C-N cross coupling on air, in addition. As a result, heterogeneous Cu catalysts, especially Cu-based nanocatalysts, seem to be essential for Cu-catalyzed methods of C-N cross coupling-based dehalogenation [136,137]. Although Cu-based C-N cross-coupling does not achieve sufficient dehalogenation efficiency [3], the produced partially halogenated aromatic amines are suitable for subsequent complete dehalogenation in aqueous solution using proved hydrodehalogenation methods accompanied by subsequent biodegradation [138–141].

**Author Contributions:** T.W. conceived, designed and wrote the paper, M.Š. provided technical support and A.Č. validated the paper. All authors have read and agreed to the published version of the manuscript.

**Funding:** This research was funded by Faculty of Chemical Technology, University of Pardubice, with the support of excellent research teams.

**Acknowledgments:** Faculty of Chemical Technology, University of Pardubice for the financial support of this review.

**Conflicts of Interest:** The authors declare no conflict of interest.

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
