# Peer review of "Copper-Catalyzed Reactions of Aryl Halides with N-Nucleophiles and Their Possible Application for Degradation of Halogenated Aromatic Contaminants"

_catalysts, doi:10.3390/catal12080911_

Round 1
Reviewer 1 Report
In the present review the authors , using the example of selected publications, describes the main achievements and directions of "post-Ulman" chemistry, namely, the copper-catalyzed amination of aryl halides. A review on this topic is a very difficult task, especially given the large amount of literature, including solid reviews, on copper-catalyzed reactions (for example, Kevin H. Shaughnessy et al. Copper-catalyzed amination of aryl and alkenyl electrophiles, 2014, Wiley; Chemical Reviews, 116 (19), pp. 12564-12649; Russian Chemical Reviews (2021), 90 (11): 1359; Catal. Sci. Technol., 2014, 4, 4169-4177, etc.), some of them are mentioned by the authors in the introduction, but this list undoubtedly needs to be expanded. The review provides a brief but quite representative list of the main protocols for copper-catalyzed amination, which makes it possible to understand what catalytic systems and conditions are common in this area. Particular attention is paid to the influence of the nature of the halogen on the choice of the catalytic system. Part 2.2 inevitably needs some expansion. In this part, in the text and in the diagrams, the influence of the nature of the amine on the reaction yield should be reflected to a greater extent, because the nature of the amine (primary, secondary, aniline, N-alkylaniline, diarylamine or azole) is a very important factor for choosing a catalytic system in the series with the nature of the aryl halide. Appropriate comments and discussions should also be added to the text. This is a major shortcoming that the authors need to address before the review can be published. There are also many technical errors: "via" should be in italics; some references in the text, names of schemes and figures are often highlighted in colors (red, blue, orange, for example, pages 20, 21, 22 and others), the font size changes. The authors should check the correspondence of the links to the text, for example, Figure 9 (p. 23, line 571) refers to reference 104, not 105. The format of the references should also be corrected. The review can be published after a major revision.
Author Response
The authors are grateful to the reviewer for their time, valuable comments and suggestions which helped to improve this manuscript.
Based on your comments:
- Introduction was completed with new information available in articles mentioned by reviewer.
- Chapter 2.2 was reorganized and supplemented with required information and additional References were added to the „References“ Chapter. Our main area of interest is to discuss possible broadly applicable protocol for dehalogenation based on C-N cross coupling. Due to this reason we focused our review on widely applicable ligands effective even for dehalogenation of non-activated chloroaromatics (class of oxalic acid amides).
- We tried to correct the text and checked the correspondence of the links to the text. We are sorry for errors. We are not native speakers.
- The format of References was unified (corrected).
Reviewer 2 Report
In this review, the author presented the recent applications of copper or copper-based compounds as a nonprecious metal catalyst in N-nucleophiles-based dehalogenation (DH) reactions of halogenated aromatic compounds (Ar-Xs). On the whole, the review is of considerable interest and well done. I recommend it to be published after a minor revision.
1. The introduction is adequate, however, the properties, microstructure and electron energy level structure for cupper as a catalyst has not been discussed and analyzed in detail, thus, it makes the review article seem shallow.
2. The Authors should also proofread their manuscript (some spelling and grammar errors).
3. - The conclusion is too long and also not targeted to the important aspects described in the manuscript; please rephrase it.
Hence, I recommend it accepted for publication after some minor revisions.
Author Response
The authors are grateful to the reviewer for their time, valuable comments and suggestions which helped to improve this manuscript.
Based on your comments:
- Additional information dealing with copper and its compounds was added and/or mentioned in Schemes (Scheme 4,5,7,8) together with links to references with detailed description (or measurements//calculations). We hope that it will be sufficient for this type of technologically focused article.
- We tried to correct the text and checked the correspondence of the links to the text. We are sorry for errors. We are not native speakers.
- The Conclusion chapter was shortened, rephrased and replenished with important aspects of our point of view (possible treatment of recalcitrant halogenated aromatics).
Round 2
Reviewer 1 Report
The authors seriously revised the manuscript and took into account all the comments. Now the review can be accepted for publication.